# Associations between Drinking Behaviors and Meaning in Life: Evidence from Primary Care Professionals in China

**DOI:** 10.3390/nu14224811

**Published:** 2022-11-14

**Authors:** Nan Yao, Zhen Wei, Yifan Wang, Long Sun

**Affiliations:** 1Centre for Health Management and Policy Research, School of Public Health, Cheeloo College of Medicine, Shandong University, Jinan 250012, China; 2NHC Key Lab of Health Economics and Policy Research, Shandong University, Jinan 250012, China

**Keywords:** meaning in life, drinking behaviors, primary care professionals, China

## Abstract

(1) Background: Although the associations between drinking behaviors and emotional problems have been supported in several previous studies, the associations between drinking behaviors and meaning in life have not been explored until now. We aimed to test the associations between drinking behaviors and meaning in life among primary care professionals, after controlling for depression. (2) Methods: In the current study, we collected 1453 valid questionnaires based on a cross-sectional design. Meaning in life, drinking behaviors, physical diseases, depression, work-related variables, and some other social-demographic variables were evaluated. (3) Results: The results support that after controlling for depression, regular milk drinking (β = 1.387, *p* = 0.026), and regular juice drinking (β = 2.316, *p* = 0.030) were associated with higher meaning in life, while regular water drinking (β = −1.448, *p* = 0.019) was negatively associated with meaning in life. In addition to this, the results showed that the older age (β = 0.098, *p* = 0.001), preventive medicine majors (β = 4.281, *p* = 0.013), working fewer days per week (β = −0.942, *p* = 0.004), licensed (assistant) technician qualification (β = 2.921, *p* = 0.036), and no depression (β = −0.203, *p* < 0.001) were positively associated with meaning in life. (4) Conclusion: This study supported the association between drinking behaviors and meaning in life, even after controlling depression. These findings imply that we can further explore this association and its mechanisms in future studies.

## 1. Background

Although meaning in life has been variously defined in different fields [1,2,3,4,5], the core contents of meaning in life contained the sense made of, and significance felt regarding the nature of one’s being and existence [6]. In other words, meaning in life could be understood as that people usually have a strong need to understand themselves and the world around them, and will seek ways to satisfy this need through a range of cognitive and behavioral activities [7,8]. Theoretically, people can find meaning in their lives when they are able to comprehend themselves and the world around them and identify what they want in life [9]. Meaning in life as an integral part of psychological functioning has been a focus of mental health studies [10,11], playing an important role in different areas such as emotions [12], health behaviors [13], and mental disorders [14,15].

More and more evidence support the theory that eating and drinking behaviors, as an important part of daily life, are associated with individuals’ physical and mental health, well-being and meaning in life. For example, eating disorders are one kind of mental disorder about people’s eating behaviors, which was highly related with people’s thoughts and emotions [16]. Previous studies had found that traditional foods can increase patients’ sense of belonging and well-being through familiar tastes and smells [17]. Another study found that vegetarians had lower self-esteem, less psychological adjustment, less meaningful life, and more negative emotions compared to semi-vegetarians and omnivores [18]. In addition to this, it has been documented that meaning in life is associated with harmful alcohol consumption [19,20] and that these associations are mediated by the value of alcohol and the mediators of trait self-control [20]. There is an association between latent toxoplasmosis and excessive alcohol consumption [21,22]. Toxoplasma infection leads to the development of central nervous system disorders that have been observed to play the greatest role in traffic road accidents, suicides, and psychiatric patients [22]. Not only that, but high chronic alcohol intake can act as a toxin to the heart and vascular system [23]. It is also true that alcoholics feel more isolated than members of most other groups, with a general dissatisfaction with most things in life [24]. All of these findings suggested to us that eating behaviors may also affect the meaning of life by influencing people’s thoughts, emotions and health.

Both cross-sectional and prospective studies have shown that lower meaning in life is associated with higher levels of depressive symptoms [12,25], and meaning in life was also one of important symptoms for depression [26,27]. There were also several experimental studies that explored the mechanism of the association between drinking behaviors and depression. For example, an experimental study demonstrated that polyphenols in pomegranate juice could provide neuroprotective functions through redox pathways, alleviating anxiety and depression-like behaviors induced by aluminum in male mice [28]. Consumption of flavonoid-rich orange juice may regulate the human gut microbiome to prevent and treat depression [29]. Olive juice contains high amounts of olive polyphenols, which act as antioxidant active substances and can improve depression in mice [30]. Other berry juices have been shown in experiments to reduce anxiety and depression-like behaviors [31,32]. It has also been found that coffee and tea consumption are associated with a reduced risk of depression, because caffeine in coffee and tea stimulates the central nervous system, enhancing dopaminergic neurotransmission and counteracting depressive states [33]. Further considering the identified associations between meaning in life and depression [12,25] suggest an association between drinking behaviors and meaning in life.

Drinking behaviors are associated with sleep quality [34], physical health [35], psychopathology [36] and aggressive behavior [37]. For example, several experimental studies, epidemiological studies [38] and clinical studies [39] have reported that flavonoids and polyphenols, which are abundant in fruits, tea and cocoa, help to reduce the risk of depression. Energy drinks and coffee consumption among early adolescents have also been shown to be cross-sectionally and longitudinally associated with similar psychopathology symptoms [36]. Soft drink consumption is a major risk factor for aggressive behavior [37] and positively associated with many chronic diseases [35]. In summary, drinking behaviors have been shown to be associated with multiple domains including mental health. However, to our knowledge, previous studies have not directly examined the association between drinking behaviors and meaning in life.

Therefore, a cross-sectional study was conducted to fill the gaps in the existing literature about the relationship between drinking behaviors and meaning in life among Chinese primary care professionals. If the association can be identified, the findings not only remind us to pay attention to people’s daily drinking habits, but also provide us with further strategies to improve meaning in life, and thus protect physical and mental health.

## 2. Methods

### 2.1. Sampling and Participants

The study recruited primary care professionals (doctor, nurse, and medical technician, etc.) based on a cross-sectional design in township health centers in Shandong Province, China. Multi-stage stratified cluster sampling was used to select the participants. First, all of the 16 metropolises in Shandong Province were classified into five groups according to 2019 GDP by metropolis. In each group, one city (metropolis) was randomly selected for interview, and five metropolises (Qingdao, Weifang, Zibo, Liaocheng, and Zaozhuang) were instructed to complete the questionnaire. Second, one county (districts) out of all counties (districts) from each metropolis was randomly selected. Third, all township health centers were selected from each county (districts) as sample units. All medical staffs working on the date of the interview were scheduled for the interview. Finally, a total of 1454 questionnaires were collected. Excluding relevant missing data resulted in a final sample of 1453 medical personnel.

### 2.2. Data Collection

This study was conducted from October to November 2020. Data were collected by the filed self-filling method. Firstly, before the survey began, all investigators received a strict training about this study. The main aims were to ensure they had sufficiently understood this study and each question in the questionnaire. Secondly, all participants were scheduled in the meeting room on the survey date. Investigators facilitated the distribution, answered the questions, and collected the questionnaires. The participants were not compensated for their participation in this study. This study was approved by the ethics committee of the school of public health, Shandong University. Informed consent was obtained after participants were informed about the purpose of the study and their right to withdraw at any time.

### 2.3. Measures

#### 2.3.1. Meaning in Life

Meaning in life was measured by using the Meaning in Life Questionnaire (MLQ), which was developed by Steger et al. (2006) [6]. The MLQ consists of two subscales: the presence of meaning in life (MLQ-P) and the search for meaning in life (MLQ-S). In this study, we focus only on the total MLQ score. The scale features 10 items with a 7-point Likert-type rating scheme, ranging from 1 (“absolutely untrue”) to 7 (“absolutely true”). Participants were asked to rate their responses to each of 10 items. The scores range from 10 to 70, and higher scores taken from the scale indicate increasing level of meaning in life. MLQ was the most widely used measure of meaning in life in a variety of populations and diverse cultures with good reliability and validity [11]. The internal consistency of this scale in the current study was Cronbach’s α 0.894.

#### 2.3.2. Drinking Behavior-Related Variables

The factors related to drinking behaviors analyzed in this study include drinking milk regularly, drinking tea regularly, drinking coffee regularly, drinking carbonated beverages regularly, drinking fruit juice regularly, drinking water regularly and drinking alcohol regularly. These behaviors are measured by some questions, for example: ‘Do you drink milk regularly?’ with response options including ‘yes’ or ‘no.’

#### 2.3.3. Social-Demographic Variables

The social-demographic variables included age, gender, marital status, academic degree, religious beliefs and family income level. Age was assessed by the date of birth of participants, and we calculated their age at the date of survey. Gender was measured by male (0) and female (1). Married status was estimated by single, married, divorced, widowed, and others. Since few subjects were single, divorced or widowed, we then categorized them into married (0) and others (1). Degree of education was measured by doctor or master, undergraduate, senior high school, junior college, technical secondary school, junior high school and below. As there were few participants who were educated higher than undergraduate and lower than senior high school, we recoded it into undergraduate and above, junior college, senior high school and below. Religious belief was measured by what religion the participants believed in, and the choices were Catholicism, Christianity, Buddhism, others, and no religion. As there were few people that had a religious belief, the religious belief was recoded into “yes” or “no.” Family income level was measured by very good, good, average, poor, very poor. We dichotomized it into “good” and “poor.” The former contained “very good” and “good,” and the latter contained “average,” “very poor” and “poor”. 

#### 2.3.4. Work-Related Variables

We also interviewed about the variables that included major, professional title, licensed (assistant) technician qualification, years of primary medical work, daily working hours, working days per week and number of night shifts per week. Majors were estimated by clinical medicine, preventive medicine, medical technology, nursing, and others. Professional titles were categorized as vice-senior and above, intermediate, and junior, and others. licensed (assistant) technician qualification was measured as “0 = no,” and “yes = 1”. 

#### 2.3.5. Physical Disease

Physical disease was assessed by one question: ‘Have you been diagnosed with a disease now?’ with response options including ‘yes’ or ‘no’.

#### 2.3.6. Depression Symptoms

Depression symptoms was measured by the 20-item Chinese version of Center for Epidemiologic Studies Depression Scale (CES-D), which was developed by Radloff (1977) [40]. The CES-D is an extremely popular scale with good reliability and validity for evaluating depression [26,41]. The scale has been widely used in various populations both domestically and internationally [42], and has been used in some literature to investigate the relationship between meaning in life and depression [43]. The responses described participants’ feelings in relation to numbers of days in the past week. Items were rated on a 4-point scale ranging from 0 (rarely or none of the time, less than 1 day) to 3 (most or all of the time, 5–7 days), with higher scores indicating a more severe depressive symptom. A cut-off score of ≥10 was used to identify the participants who had significant depressive symptoms. The internal consistency of this scale in the current study was Cronbach’s α 0.884.

### 2.4. Statistical Analysis

SPSS 21.0 for Windows (IBM, Armonk, NY, USA) was used for data analysis. we used frequency and percentage to describe the demographic characteristics of the participants. T tests or one-way ANOVA was performed to analyze the factors associated with meaning in life among primary care professionals. Multiple linear stepwise regression was conducted to examine the factors associated with meaning in life, and dummy variables were set for the variables with multi-classification. All tests were two-tailed and a *p*-value < 0.05 was considered statistically significant.

## 3. Results

In the current study, we interviewed a total of 1453 primary care professionals in Shandong province, China. Table 1 gives the characteristics of the participants and the mean ± SD of MLQ scores. The results showed that meaning in life was associated with age (F = 5.817, *p* = 0.003), licensed (assistant) technician qualification (t = 2.273, *p* = 0.023), working days per week (t = 2.192, *p* = 0.029), and depression (t = −7.242, *p* < 0.001).

In Table 2, we analyzed the association between different drinking behaviors and meaning in life among primary care professionals. In Figure 1, we further used box plots to compare the scores of meaningfulness of life for different drinking behaviors. Results demonstrated that drinking milk regularly (t = 2.268, *p* = 0.023), drinking tea regularly (t = 2.077, *p* = 0.038), drinking fruit juice regularly (t = 2.474, *p* = 0.013), and drinking water regularly (t = −2.789, *p* = 0.005) were significantly associated with meaning in life. Among them, drinking milk, juice and tea regularly had high meaning in life, while drinking water regularly had low life significance.

Finally, we conducted multiple linear stepwise regression to identify the factors associated meaning in life in Table 3. After controlling for depression, the results supported that the older age (β = 0.098, *p* = 0.001), preventive medicine majors (β = 4.281, *p* = 0.013), working fewer days per week (β = −0.942, *p* = 0.004), licensed (assistant) technician qualification (β = 2.921, *p* = 0.036), no depression (β = −0.203, *p* < 0.001), not drinking water regularly (β = −1.448, *p* = 0.019), drinking milk regularly (β = 1.387, *p* = 0.026), and drinking fruit juice regularly (β = 2.316, *p* = 0.030) were associated with higher levels of meaning in life.

## 4. Discussion

In the current study, we mainly analyzed the association between drinking behaviors and meaning in life among Chinese primary care professionals. To our knowledge, this is the first study to examine the relationship between drinking behaviors and meaning in life. Results indicated drinking milk and fruit juice regularly associated with higher levels of meaning in life. However, drinking water regularly was associated with lower meaning in life. We also analyzed other variables in this study, and the results supported that age, preventive medicine majors, working days per week, licensed (assistant) technician qualification, and depression were also associated with meaning in life. 

First, we found that drinking milk regularly was positively associated with meaning in life. One possible explanation for this relationship may be that people who drink milk regularly have a lower risk of developing depression. Previously, a study based on US adults demonstrated that protein intake from milk and dairy products may reduce the risk of depressive symptoms [44]. Furthermore, phospholipids in milk may also have a neuroprotective effect and reduce the risk of depression [45]. Second, we found that people who regularly drank fruit juices also had higher meaning in life. As we mentioned in the introduction section, the polyphenols, flavonoids, and other substances contained in different types of fruit juices can affect the body’s mechanisms in different ways, thus reducing the risk of depression [28,29,30,31,32]. Depression is known to be negatively associated with meaning in life, which has been confirmed in previous studies [12,25]. Therefore, regular milk and juice consumption may improve the meaningfulness of life by reducing depression. However, there is no literature to prove whether depression plays a role in this, and further studies are needed to verify.

In the present investigation, drinking water regularly is another important factor that merits attention. The results showed that people who drink water regularly have lower meaning in life. Previous studies have demonstrated that meaning in life is closely related to mood and cognition. Changes in hydration status can affect a person’s mood and cognition [46], and most of these studies have focused on the effects of dehydration [47] and excessive water consumption states [48] on the human body. Thirst and the need for fluid regulation may motivate the intake of water. In addition to this, boredom as well as sadness are the main determinants of excessive fluid consumption. Therefore, these individuals who drink water regularly may drink excessive amounts of water in their daily life or always be in a dehydrated state of thirst, and the occurrence of boredom, anxiety, and depression also contribute to their frequent drinking behavior. All of these reasons can lead to emotional and cognitive dissonance and thus make life less meaningful.

In the univariate analysis of this study, we also found a trend that regular tea and coffee consumption was associated with higher meaning in life, and regular carbonated beverage and alcohol consumption was associated with lower meaning in life. Cumulative evidence from clinical and epidemiological studies has linked tea and coffee consumption to a variety of health benefits [49,50]. It has been demonstrated that alcohol abuse can be hazardous to health, such as toxoplasmosis infection [21] and cardiovascular disease development [23], which may be risk factors of meaning in life. In addition to this, excessive alcohol consumption has been shown to be negatively associated with meaning in life [20,51], which is consistent with the present study. People with low meaning in life are more likely to abuse alcohol, which serves as an outlet for cumulative stress that stems from the pursuit of professional perfection and serious emotional deficits in the personality domain [52]. The motivational value of alcohol increases the emotional satisfaction that individuals derive from life [19]. Regular consumption of carbonated beverages has also been shown to be associated with obesity and other adverse health outcomes [53]. A recent study suggests that meaning in life is associated with health behaviors in which risk perception plays a mediating role [54]. Thus, this implies to us that people with high meaning in life may be likely to choose more expensive beverages, or perhaps beverages that have a reputation of being healthful.

Another possible explanation for this relationship between drinking behaviors and meaning in life may be that the food and drink that we consume have an impact on several functional areas of the brain such as neuronal and synaptic plasticity, which affects a person’s cognitive, emotional, and psychological well-being [55]. Previous studies have found that meaning in life was negatively associated with eating disorders and a negative attitude toward food [14]. In addition to this, people with eating disorders experience low quality of life and mood disorders [56], which are negatively correlated with meaning in life.

In this study, we also found older people had a stronger sense of the meaning in life than younger people. The relationship between age and the meaning in life has been reviewed in several previous reports, but the results were not always consistent between studies [57,58,59]. Some of these studies suggested that scores for meaning in life increased with age [60,61], which is consistent with the findings of this study. This result could be supported by Erik Erikson’s theory of psychosocial development [62], which states that people typically go through a series of stages centered on social and emotional development. Specifically, people’s understanding and perception of their identity evolves as they age and meaning in life increases in the process [57]. Some studies found no age differences in meaning scores [63]. However, some current literature suggests that there should be a nonlinear relationship between age and meaning in life, e.g., J-shaped [57], U-shaped [58], etc.

Depression was also an important factor associated with meaning in life. The results of this study also suggested that depression is negatively related to meaning in life [12,25]. It is consistent with the finding in previous studies. Depression reduces the positive events [64] and affect [65] used to create or maintain meaning. Previous studies demonstrated that older adults with depressive symptoms may be particularly prone to perceive life as meaningless [66]. 

We also analyzed the relationship between work-related variables and meaning in life. Licensed (assistant) technician status means that the person’s ability to practice is recognized by the public and the state, and that the person can work for a better institution or organization [67]. The existing literature suggested that meaning in work was positively correlated with meaning in life and that work was a strong source of meaning in life [68]. Our results also supported that working days per week were negatively associated with meaning in life. Long working hours have been shown to be associated with poorer mental health and increased levels of anxiety and depressive symptoms [69]. The same results have been identified by many previous studies worldwide [70,71,72]. All these negative symptoms were risk factors for low meaning in life. In addition, the high sense of meaning in life in preventive medicine relative to other medical specialties may be explained by an enhanced sense of professional identity and a sense of personal value. Politics, economics, and policy have long undervalued and underfunded public health and population health, and the 2020 New Coronary Pneumonia pandemic prompted medicine and public health to prioritize prevention and wellness over treatment and disease [73]. An overview and analytical study of drug program costs in Poland from 2015 to 2018 also shows that increased annual drug program spending in Poland has improved patient access to oncology and non-oncology treatments in the Polish healthcare system [74].

There are also several limitations that should be considered when we are interpreting these results. First, this was a cross-sectional study, and causal inferences between the variables are difficult to prove. Thus, it would be beneficial to use longitudinal data to confirm causal relationships in future studies. Second, the data was collected from primary care professionals in Shandong Province, China, and applicability of the results to other populations is not assured. Third, drinking behaviors involved numerous factors such as frequency of consumption, type of drink and information on exact elemental content, so the questions assessed in this study were relatively simple. Last, the percentages of drinking coffee, carbonated beverages, fruit juice, and alcohol were at a low level. The one-sided trend may also cause some bias for the results.

## 5. Conclusions

Despite these limitations, the study is the first to report the association between drinking behaviors and meaning in life among Chinese primary care professionals. These findings should be paid more attention because meaning in life has emerged as an important focus of study in medical research and has been shown to be an important factor in physical and mental health [75,76]. This indicates that we can further explore this association and its mechanisms in future studies.

## Figures and Tables

**Figure 1 nutrients-14-04811-f001:**
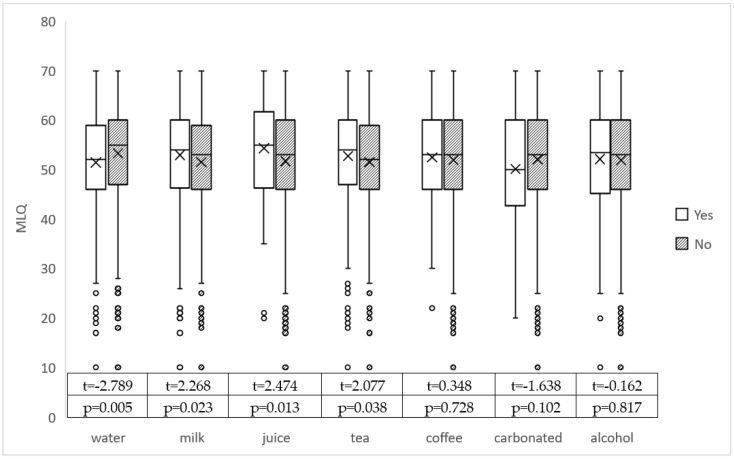
Box plot of the association between drinking behaviors and meaning in life.

**Table 1 nutrients-14-04811-t001:** Sample characteristics and single analysis of MLQ among the GPs in Shandong, China.

Variable	N (%)	MLQ (M ± SD)	*t/F*	*p*
Observations	1453	-	-	-
Age			5.817	0.003
18-	540 (37.2)	50.77 ± 10.85		
35-	760 (52.3)	52.43 ± 10.46		
50-	153 (10.5)	53.51 ± 10.02		
Gender			1.241	0.215
Male	409 (28.1)	52.48 ± 51.71		
Female	1044 (71.9)	51.71 ± 10.36		
Marital status			1.591	0.112
Married	1155 (79.5)	52.15 ± 10.57		
Others	298 (20.5)	51.06 ± 10.69		
Academic Degree			0.103	0.902
Undergraduate and above	751 (51.7)	51.92 ± 10.29		
Junior college	523 (36.0)	52.04 ± 10.74		
Senior high school and below	179 (12.3)	51.63 ± 11.49		
Religious beliefs			−1.185	0.236
Yes	49 (3.4)	50.16 ± 11.95		
No	1404 (96.6)	51.99 ± 10.55		
Years of primary medical work			−1.450	0.147
≤10	720 (49.6)	51.52 ± 10.92		
>10	733 (50.4)	52.33 ± 10.26		
Family income level			1.802	0.072
Good	149 (10.3)	53.41 ± 10.73		
Poor	1304 (89.7)	51.76 ± 10.57		
Major			1.745	0.138
Clinical medicine	397 (27.3)	51.56 ± 11.04		
Preventive medicine	37 (2.5)	56.30 ± 9.41		
Medical technology	112 (7.7)	51.48 ± 9.75		
Nursing	534 (36.8)	51.94 ± 10.84		
Others	373 (25.7)	52.00 ± 10.06		
Professional title			1.552	0.199
Vice-Senior and above	71 (4.9)	53.51 ± 8.605		
Intermediate	380 (26.2)	51.77 ± 10.00		
Junior or assistant	578 (39.8)	52.37 ± 11.03		
Others	424 (29.2)	51.20 ± 10.80		
Licensed (assistant) technician qualification			2.273	0.023
Yes	57 (3.9)	55.05 ± 9.29		
No	1396 (96.1)	51.80 ± 10.63		
Daily working hours			1.825	0.068
≤8	1226 (84.4)	52.15 ± 10.40		
>8	227 (15.6)	50.75 ± 11.55		
Working days per week			2.192	0.029
≤5	291 (20.0)	53.14 ± 10.67		
>5	1162 (80.0)	51.62 ± 10.56		
Number of night shifts per week			2.174	0.089
0	863 (59.4)	51.62 ± 10.41		
1	229 (15.8)	52.55 ± 10.70		
2	221 (15.2)	53.23 ± 10.39		
≥3	140 (9.6)	50.78 ± 11.71		
Physical disease			−1.750	0.080
Yes	177 (12.2)	50.62 ± 10.21		
No	1276 (87.8)	52.11 ± 10.64		
Depression			−7.242	<0.001
Yes	412 (28.4)	48.63 ± 11.26		
No	1041 (71.6)	53.23 ± 10.03		

Note: SD = standard deviation; MLQ = Meaning in Life Questionnaire.

**Table 2 nutrients-14-04811-t002:** The association between drinking behaviors and MLQ among the GPs in Shandong, China.

Drinking Behavior	N (%)	MLQ (M ± SD)	*t*	*p*
Drinking milk regularly			2.268	0.023
Yes	400 (27.5)	52.95 ± 11.04		
No	1053 (72.5)	51.54 ± 10.40		
Drinking tea regularly			2.077	0.038
Yes	462 (31.8)	52.77 ± 10.80		
No	991 (68.2)	51.53 ± 10.48		
Drinking coffee regularly			0.348	0.728
Yes	50 (3.4)	52.44 ± 11.18		
No	1403 (96.6)	51.91 ± 10.58		
Drinking carbonated beverages regularly			−1.638	0.102
Yes	90 (6.2)	50.16 ± 11.27		
No	1363 (93.8)	52.04 ± 10.55		
Drinking fruit juice regularly			2.474	0.013
Yes	108 (7.4)	54.35 ± 10.70		
No	1345 (92.6)	51.73 ± 10.57		
Drinking water regularly			−2.789	0.005
Yes	1058 (72.8)	51.43 ± 10.20		
No	395 (27.2)	53.27 ± 11.51		
Drinking alcohol regularly			−0.162	0.817
Yes	72 (5.0)	51.92 ± 10.57		
No	1381 (95.0)	52.13 ± 11.26		

Note: SD = standard deviation; MLQ = Meaning in Life Questionnaire.

**Table 3 nutrients-14-04811-t003:** Multiple linear stepwise regression analysis of MLQ among the GPs in Shandong, China.

Model	*β*	95% CI	*p*
Constant	56.448	51.828, 61.068	<0.001
Age	0.098	0.038, 0.158	0.001
Preventive Medicine	4.281	0.896, 7.666	0.013
Working days per week	−0.942	−1.591, −0.293	0.004
Licensed assistant technician qualification	2.921	0.191, 5.650	0.036
Depression	−0.203	−0.258, −0.147	<0.001
Drinking water regularly	−1.448	−2.659, −0.236	0.019
Drinking milk regularly	1.387	0.167, 2.607	0.026
Drinking fruit juice regularly	2.316	0.226, 4.406	0.030

Note: Adjust R^2^ = 0.06; CI = confidence interval; β = partial regression coefficient.

## Data Availability

The data sets used and/or analyzed during the current study are available from the corresponding author on reasonable request.

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
