# Peer review of "Associations between Drinking Behaviors and Meaning in Life: Evidence from Primary Care Professionals in China"

_nutrients, 2022, doi:10.3390/nu14224811_

Round 1

Reviewer 1 Report

Review: Yao et al: “Associations Between Drinking Behaviors and Meaning in Life: Evidences from Primary Care Professionals in China

This submission is a poor fit for MDPI Nutrients. I recommend that you transfer it to MDPI Psych.

The results show a statistical association between responses to certain written questions and scores on a meaning in life survey. They also show an association between a dichotomized score on a depression survey and the meaning in life survey. The authors offer several speculations on how drinking certain beverages could lead to less depression and higher MIL. To give a fair interpretation they should also explore the possibility that high MIL could lead to selection of sweeter, more expensive beverages, or perhaps beverages that have a reputation of being healthful.

There may be some issues with the disproportion in responses. Only 3.4% of respondents claimed to drink coffee regularly, 6.2% drank carbonated beverages, and 7.4% drank fruit juice. The authors should comment on whether such a one-sided trend in these variables could affect the reliability of the analysis.

The conclusion is seriously deficient. This study does not report the effect of drinking behavior on MIL. It could just as easily be interpreted as showing the effect of MIL on drinking behavior, or the effect of a third variable, such as depression, on both.

The attached file has my handwritten edits for pages 1-9.. There should be a single space between the brackets containing an in-line citation, and the preceding text: line 36 “behaviors [13]” not “behaviors[13]”.

In the references section, abbreviations in journal titles should have a period: “J. Gerontol.” Not “J Gerontol”

Author Response

请参阅附件。

Reviewer 2 Report

Although this is an interesting paper, there are some points that need major revision:

1. It is not clear how the hypothesis is formed based on the selected literature in the introduction. A major revision of this section is needed. The discussion is brief and not adequately supported by scientific literature. For example, why is alcohol not mentioned? Even if excluded, it must be discussed. Please see some indicative relevant references to be discussed below:

Copeland, A., Jones, A., & Field, M. (2020). The association between meaning in life and harmful drinking is mediated by individual differences in self-control and alcohol value. Addictive behaviors reports11, 100258.

Jaffe, A. E., Kumar, S. A., Hultgren, B. A., Smith-LeCavalier, K. N., Garcia, T. A., Canning, J. R., & Larimer, M. E. (2022). Meaning in life and stress-related drinking: A multicohort study of college students during the COVID-19 pandemic. Addictive behaviors129, 107281.

Ivanova, D., & Giannouli, V. (2017). Lesch type III alcoholism in Bulgarian women: Implications and recommendations for psychotherapy. International Journal of Caring Sciences10(3), 1569-1576.

Palfai, T. P., & Weafer, J. (2006). College student drinking and meaning in the pursuit of life goals. Psychology of Addictive Behaviors20(2), 131.

2. More details are necessary in the methodology regarding the recruitment of the participants. The tools are well-described, but why CESD is included instead of other questionnaires? Why are these variables included instead of others? There is no literature support behind the hypothesis.

3. The statistics are basic. More complex analyses are needed with all these variables.

Reviewer 3 Report

The article by Yau et al. Associations Between Drinking Behaviors and Meaning in Life: Evidences from Primary Care Professionals in China" covers a potentially interesting and emerging topic related to the drinking behaviors and emotion problems. In this sense, this remains to be potentially interesting for the Nutrients readers. I regard the main point of this paper as highly attractive as well as the results are clearly presented. The text does not contain any major errors, therefore I have some minor comments and

recommendations:

1. There is a need to provide slightly more expanded introduction regarding social costs of drinking abuse

2. The figure summarizing and clarifying above statistics should be added

3. Following references should be added and properly cited within the main text:

Samojłowicz D, Twarowska-Małczyńska J, Borowska-Solonynko A, Poniatowski ŁA, Sharma N, Olczak M. Presence of Toxoplasma gondii infection in brain as a potential cause of risky behavior: a report of 102 autopsy cases. Eur J Clin Microbiol Infect Dis. 2019 Feb;38(2):305-317. doi: 10.1007/s10096-018-3427-z.

Marcos A, Serra-Majem L, Pérez-Jiménez F, Pascual V, Tinahones FJ, Estruch R. Moderate Consumption of Beer and Its Effects on Cardiovascular and Metabolic Health: An Updated Review of Recent Scientific Evidence. Nutrients. 2021 Mar 9;13(3):879. doi: 

Mela A, Poniatowski ŁA, Drop B, Furtak-Niczyporuk M, Jaroszyński J, Wrona W, Staniszewska A, Dąbrowski J, Czajka A, Jagielska B, Wojciechowska M, Niewada M. Overview and Analysis of the Cost of Drug Programs in Poland: Public Payer Expenditures and Coverage of Cancer and Non-Neoplastic Diseases Related Drug Therapies from 2015-2018 Years. Front Pharmacol. 2020 Aug 14;11:1123. doi: 10.3389/fphar.2020.01123. 

Akerlind I, Hörnquist JO. Loneliness and alcohol abuse: a review of evidences of an interplay. Soc Sci Med. 1992 Feb;34(4):405-14. 

4. In some places the use of English could be improved on.

Completing this gaps will have an impact on the understanding the aim of the study and, from my point of view, is absolutely necessary.

Round 2

Reviewer 1 Report

In my original review, I suggested that the subject matter is not a good match for MDPI Nutrition. It would be better placed in MDPI Psych.

The authors did not adopt most of my edits (attached again herewith). These are not mandatory. The paper is understandable as is, but the presentation would be more professional if my edits were adopted.

I am pleased that the interpretation of the results was clarified, and that the non-correlation with alcohol consumption was included.
